# A System to Support the Transparency of Consumer Credit Offers

**Bożena Frączek** 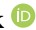

Department of Banking and Financial Markets, University of Economics in Katowice, 40-750 Katowice, Poland;
b.fraczek@ue.katowice.pl

**Abstract:** The development of financial markets and the low level of financial literacy does not facilitate consumer protection. A significant problem is the lack of information or unclear information regarding financial offers, including consumer credit. Financial protection for consumers can be increased by using systems that support consumers faced with a lack of transparency of consumer credit offers. The theoretical objective of the research is to identify the completeness of information allowing for verifying the annual percentage rate (APR) in the consumer credit offers presented and compared on websites of financial intermediaries and banks as well as the identify of APR calculation methods. After analysis of different credit offers (document analysis), a prototype of the system regarding credit costs measured at the appropriate interest rates (APR and AER) calculated in the correct manner was created. This may facilitate practices for informing consumers about the characteristics of consumer credit and at the same time support the implementation of the concept of responsible lending. The system developed uses a computer to simulate human thinking and to augment it with artificial intelligence. It facilitates the elimination the behavioral biases during the taking of financial decisions, which are the result of a low level of financial literacy.

**Keywords:** consumer credit/loan; deficit of financial information; financial consumer protection; annual percentage rate, APR; annual effective rate, AER; periodic rate

## 1. Introduction

Adaptation of the promoted idea of responsible lending (OECD 2019; Wilson 2013) to real lending processes require meeting many conditions by different entities: loan providers and intermediaries with their good practice realization, responsible credit policy makers, as well as educated and conscious credit consumers (Ramsay and Williams 2020; Cherednychenko and Meindertsma 2019).

The potential borrowers can receive loan offers from multiple lenders: directly from the banks but also from private lenders and different intermediaries (Ivashina et al. 2020; Bertsch and Rosenvinge 2019; Bruckner 2018; Gutiérrez-Nieto et al. 2011). The online loan marketplace, represented by numerous of online platforms (e.g., LendingTree) empowers consumers as they comparison-shop across a full suite of loan and credit-based offerings. Platforms connect consumers with multiple lenders as well as offer an array of online tools and information to help consumers find the best loan. The results of research confirm that digitalization in the financial sector may facilitate providing information upfront in a clear and comparable way (Suter et al. 2019, p. 7), but on the other hand, some practices in information providing may be problematic (Jagtiani and John 2018).

Consumers have the option to use price comparison websites. However, providing financial information is inherently complex, making it challenging for consumers to assess the risks and charges involved as well as compare across providers. Among many unfair practices, there is price obfuscation, reducing the customer's ability to fully understand the price and therefore compare prices. The results of the latest research confirm that online comparison tools are seen by most users (90%) as the quickest

way to compare prices, and more than half stated that they usually purchased the cheapest product displayed on the website. Most respondents were rather negative toward price comparison websites, stating that these tools are not independent and impartial (Suter et al. 2019, pp. 58–59).

With regard to credit consumers, results of conducted research confirm that credit consumers do not always pay enough attention to loans terms and conditions or/and they do not fully understand the cost implication of a loan. In addition, the existence of deceptive advertising encouraging the use of financial offers (e.g., financial calculators available on the internet) is a very serious problem of contemporary financial markets, where ongoing progress in expanding financial access can be observed (Frączek and Mitręga-Niestrój 2014; World Bank 2012, 2015, 2018). A failure to understand the terms and conditions of financial products on the part of individual financial consumers (Ardic et al. 2011) can be exacerbated by unclear and nontransparent disclosure of prices and fees. Despite the existing requirements and still extended regulations on retail financial products and consumer credit, there is still evidence that consumers do not fully understand the prices and fees associated with these products (OECD 2019, pp. 25–26). Credit consumers may suffer from cognitive limitations that make the comparison of products and prices harder, or they may be prone to a wide range of behavioral biases (Greenberg and Hershfield 2018; Zinkhan and Braunsberger 2004). Sometimes, consumers focus only on the installments, rather than focus on the total cost of credit or APR, which may be not understandable for them (OECD 2019, pp. 23–25). Due to a lack of basic financial knowledge, they cannot verify the correctness of APR calculation according to standards/rules.

Under consumer protection, especially in relation to consumer credit transactions, many recommendations are developed. They usually address supervision and financial education issues related to (among others) the equitable and fair treatment of consumers as well as disclosure and transparency and the protection of consumer assets against fraud and misuse (OECD 2018). The research conducted and actions taken to date include the promotion of Good Practices on Financial Education and Awareness Relating to Credit (OECD 2009) and Consumer Protection in the field of consumer credit (OECD 2018) as well as guidance to supervisors on consumer credit, among other areas (FinCoNet 2019). New work streams are developed in the field of product culture and governance and on financial advertising (OECD 2019, p. 9).

The described aspects in three groups of entities (lenders, credit consumers, and consumer credit policy makers) involved (directly or indirectly) in the lending process tend to underline the problem of the deficit of information transparency. In the financial market, the problem of a lack of information transparency concerns the lack of transparency in financial regulations (Kaufmann and Weber 2010), the activities of opaque financial institutions (e.g., hedge funds), opaque financial assets (e.g., structured products) (Sato 2014), as well as a lack of transparency of information regarding the financial terms of offers issued by financial institutions. The reasons for and implications of opacity in financial markets are not fully understood. On the one hand, opacity may be seen as a side effect associated with sophisticated financial techniques, for example to achieve better risk sharing (Kaufmann and Weber 2010), but on the other hand; the lack of transparency in financial market exists for the sole purpose of obfuscating true payments regarding financial instruments, which causes losses among unaware financial consumers (World Bank Development Research Group and GFLEC 2015).

In this paper, the lack of transparency is considered from the point of view of the emergence of the danger of obfuscating true payments in the form of income and costs related to basic financial offers.

An unclear definition of the terms of financial offers may be seen as a strategic tool for exploiting less-informed consumers (Carlin and Manso 2011). In many cases, this opacity allows lenders to collect higher fees by manipulating how the interest rates (e.g., annual percentage rate, APR) are presented in their offers.

There has been research on APRs, whose results may be used in the process of taking the loan/credit. Research on Annual Percentage Rate (APR) usually regards the average offered APRs, average loan amount (with given scores in given periods), and differences between APRs on personal loans in

comparison to collateralized debt. The result of other research is a database with APR information on some or all of the significant lenders in different countries (Rosenberg et al. 2013). Sometimes, studies in the literature present the relationship between the APR, the term of the loan, and the monthly repayments and overall financial cost and analyze the effects of these drivers on determining the financial decision about borrowing money. Many studies regularly examine prevailing interest rates and analyze the range of APRs (the spread is usually large) (Lunn et al. 2016, pp. 1–14).

The research available in the literature on APR includes neither studies verifying the completeness of the information that would allow the verification of the APR provided nor research verifying whether the annual percentage rate (APR) is always calculated according to the same standards. Moreover, there is no proposal for simple tools or systems to fill gaps in information and to support unsophisticated, uneducated, and unaware financial consumers.

The identified research gap allowed for the formulation of the research objectives.

The theoretical objective of the research is to identify the completeness of information allowing verification of the APR in the consumer credit offers presented and compared on websites of financial intermediaries and identify APR calculation methods. The empirical objective is the development of conclusions and implications that help to develop a system to support the increase the transparency of consumer credit offers regarding credit costs measured at the appropriate interest rates (annual percentage rate, APR and annual effective rate, AER) calculated in the correct way and presentation of a prototype for such a system.

## 2. Methods

The objectives are detailed in the form of research questions:

1. Is every financial offer regarding consumer credit presented in a transparent and clear way?
2. Does the presentation of the costs of every credit offer use standardized parameter APR calculated in the same way, which makes it possible for credit offers to be properly compared?
3. Is it possible to develop a system to support and increase the transparency of consumer credit, which also increases consumer protection?

The research consisted of document analysis, which is a form of qualitative research that analyzes documentary evidence and answers specific research questions (Frey 2018).

This was a non-probability research sample due to the intention to demonstrate any case of opacity of information regarding cost of credit and various way of calculating the APR in credit offers. In the research, document analysis was undertaken, because documents constitute the main evidence in an enquiry to support observational data. Documents used in the qualitative research have a virtual version of genuine consumer credit offers, presented by banks and intermediaries presented at the websites of banks and intermediaries.

Document analysis involves analyzing and interpreting data generated from the examination of documents, and this is looking for supposed factual evidence or confirmation allowing research questions to be answered. Close scrutiny of documents is necessary when the document has errors or has internal style or content inconsistencies (Platt 1981).

The ultimate purpose of examining documents is to arrive at an understanding of the meaning and significance of what the document contains (Scott 1990, p. 28).

To answer the 1st and 2nd research question, an analysis of a range of banking was conducted that enables the principal costs and fee payment methods to be identified. The analysis was preceded by presenting the annual percentage rate as the important criterion of credit decision making.

To answer the 3rd research question, a system supporting and increasing the transparency of consumer credit offers will be developed. The idea of the system is to present and process information on credit offers that is available but incomplete in order to provide variant solutions that may be helpful in informed credit decision-making. The final result will be various sets of the most important parameters,

verifying the information under credit offers. In addition, the results of the developed supporting system can suggest questions to be asked by financial consumers before making financial decisions.

## 3. Annual Percentage Rate as the Important Criterion of Credit Decision Making

Borrowers are interested in the terms of qualifying for a loan, and the ways and restrictions on how a borrower may use borrowed money so that he/she will not get into legal trouble, but from the financial point of view, they should be interested in the costs associated with credit/loans. In addition to interest, consumer credit may include various charges such as fees and levies (e.g., commission, origination fee, appraisal fee, survey fee, broker fee).

The reason for the wide range of fees that constitute an additional cost of consumer credit is a low interest rate as well as competition on the credit/lending market, resulting in lenders' novel business practices aimed at finding new revenue sources (Cherednychenko and Meindertsma 2019).

Most often, potential borrowers have the choice of a fixed-rate loan with a prime market interest rate supplemented by an additional loan origination fee or the same loan offered at above the prime rate with no origination fee. The additional fees may be taken as a one-time fee, or the fee may be divided and paid along with each payment (with or without interest). However, a significant problem is that fees are not disclosed clearly or prominently. In addition, the general public does not always understand the costs associated with a loan/credit due to their low level of financial literacy. Evaluating the total cost of credit and making comparisons for borrowing money in the modern credit environment can be a daunting or even impossible task for those uneducated in finance and/or unaware financial consumers. This facilitates abusive lending practices, which have been described in the literature on numerous occasions (Burrell 2016; Ozbek et al. 2012; Renuart 2008).

One of the most important but problematic elements of consumer credit offers is the annual percentage rate (APR), which allows loan offers to be compared. The basic problem with these offers is that the annual percentage rate (APR) is usually presented without distinction between the APR in the nominal dimension—i.e., without recognizing the effect of intraperiod compounding, and the AER in the real dimension—i.e., AER annualizes the periodic effective rate by incorporating intraperiod compounding. The starting point and common building block for calculating the APR and AER is the periodic effective rate (r). The basic equation for the periodic effective rate (r) that establishes the APR and AER formulas is standardized and is presented in the annexes to many financial regulations all over the world (Vicknair and Wright 2015). In practice, the periodic effective rate (r) is usually calculated by using the concept of internal rate of return (IRR) i.e., the RATE function (RATE (nper, pmt, pv, [fv], [type], [guess])). Most real credit offers quote the APR, but while in some cases it is actually the APR, in others, it may in fact be the AER.

## 4. Analysis and Discussion

Different types of credit/loans have different terms, especially with regard to the costs. The idea of a system supporting the transparency of consumer credit offers is to facilitate the consumer (borrower) decision-making process through an improvement of information disclosure aimed at redressing information asymmetries between credit institutions and credit intermediaries and consumers. An analysis of a range of banking offers allows for the principal costs and fee payment methods to be identified (Frączek 2020):

- 1st case: Interest only, without fees. The borrower receives the full amount of the loan/credit.
- 2nd case: Interest and additional fee, wherein the full fee is charged at the beginning of the loan process. The borrower receives the full amount of the loan/credit minus the fee. In this case, it is possible to calculate the equivalent PMT (amount of repayment) for when the borrower receives the full amount of payment): a modified Case 2.
- 3rd case: Interest and additional fee, wherein the fee is added to the amount of credit and repaid (with additional interest). The borrower receives the full amount of the loan/credit.

- 4th case: Interest and additional fee, wherein the fee is added to the amount of the credit and repaid (without additional interest). The borrower receives the full amount of the loan/credit.

The cases listed above result in different PMT, APR, and AER.

Consumer credit/loan offers usually include the annual rate, the number of monthly repayments, the PMT, and information on the amount of the additional fee (in percentage or nominal format), as well as the APR. Usually, there is no information on how the fee is charged, although this is important information that determines the final cost of the loan (APR). Tables 1 and 2 present ten examples of consumer credit offers provided by two different intermediaries via different websites (I and II) and two banks. Under document analysis, the author should analyze each example and discuss the results and how they can be interpreted from the perspective of previous studies and the research questions. The findings and their implications should be discussed in the broadest context possible. Future research directions may also be highlighted.

**Table 1.** Actual consumer credit offers: eight examples. Terms of credit: amount of credit: 20,000 PLN, 48 monthly repayments (Constant PMT—Annuities).

| | Consumer Credit Offer No. 1 | | | |
|---|---|---|---|---|
| | PMT * [PLN] | APR ** | Interest rate | Fee |
| Santander offer I | 487 | 8.13% | 7.84% | 0% |
| | Consumer Credit Offer No. 2 | | | |
| | PMT * [PLN] | APR ** | Interest rate | Fee |
| Credit Agricole offer II | 511.78 | 10.47% | 9.99% | 0% |
| | Consumer Credit Offer No. 3 | | | |
| | PMT * [PLN] | APR ** | Interest rate | Fee |
| CitiHandlowy offer I | no data | 9.34% | 7.49% | 2.90% |
| | Consumer Credit Offer No. 4 | | | |
| | PMT * [PLN] | APR ** | Interest rate | Fee |
| CitiHandlowy offer II | 500.75 | 9.32% | 7.49% | 2.90% |
| | Consumer Credit Offer No. 5 | | | |
| | PMT * [PLN] | APR ** | Interest rate | Fee |
| Alior Bank offer I | 536 | 13.76% | 7.49% | 10.90% |
| | Consumer Credit Offer No. 6 | | | |
| | PMT * [PLN] | APR ** | Interest rate | Fee |
| Alior Bank offer II | 511.1 | 10.40% | 9.50% | 6.90% |
| | Consumer Credit Offer No. 7 | | | |
| | PMT * [PLN] | APR ** | Interest rate | Fee |
| PNB Paribas offer I | 538 | 13.92% | 9.99% | 6.00% |
| | Consumer Credit Offer No. 8 | | | |
| | PMT * [PLN] | APR ** | Interest rate | Fee |
| PNB Paribas offer II | 514 | 10.70% | 6.99% | 9.90% |

* PMT: amount of periodical (monthly) credit/loan repayment. ** APR: annual percentage rate. Source: https://www.bankier.pl/smart/kredyty-gotowkowe (I), https://www.kalkulator.pl/kredyt-gotowkowy/ (II), accessed: 20th of April 2020.

**Table 2.** Actual consumer credit offers: two examples. Terms of credit: amount of credit: 20,000 GBP, 48 monthly repayments (Constant PMT—Annuities).

| | Consumer Credit Offer No. 9 | | | |
|---|---|---|---|---|
| HSBC UK | PMT * [GBP] | APR ** | Interest rate | Fee |
| | 474.32 | 6.7% | 6.7% | 0% |
| | Consumer Credit Offer No. 10 | | | |
| Barclays Bank | PMT * [GBP] | APR ** | Interest rate | Fee |
| | 475.19 | 6.8% | 6.8% | 0% |

\* PMT: amount of periodical (monthly) credit/loan repayment.   \*\* APR: annual percentage rate.   Source: https://www.hsbc.co.uk/loans/products/personal/, https://www.barclays.co.uk/loans/calculator/#1, accessed: 15th of November 2020.

The offers presented in Table 1 do not provide information about how the fee is charged, and at the same time, there is no detailed information regarding the way the PMT is calculated. Usually, when credit consumers are making a financial decision, they do not understand the differences and take into account the APR, which should by definition take into account all the costs of the credit.

The considered problem does not only regard the intermediaries. The cases of Barclays Bank and HSNC in the UK confirm that there is also opacity in financial terms of consumer loans/credits; see Table 2.

The results of the analysis presented in Tables 1 and 2 provide an answer to research question no 1. Unfortunately, there are financial offers regarding consumer credit on offer in the financial market, which are not presented in a transparent and clear way.

It is at this stage of document analysis that it would be extremely useful to employ a system developed to support the transparency of consumer credit offers, as described by the algorithm presented in Figure 1.

A system that supports the transparency of consumer credit offers may provide results on PMT, APR, and AER both for cases when credit consumers have information about the details of the costs (the ways all costs are charged), as well as for all cases (four cases are presented on page 4) when there is no detailed information on the ways the costs are charged. Under consideration is a second option (for all cases) where the system supporting the transparency of consumer credit helps to set in order the necessary calculations (mainly PMT, APR, and AER), provide proper results for these parameters, and at the same time may reveal incorrect or unexplained PMT, APR, and AER of the financial offers analyzed. Document analysis based on the particular examples presented in Tables 1 and 2, and the usage of the system supporting the transparency of consumer credit offers, are presented in Table 3.

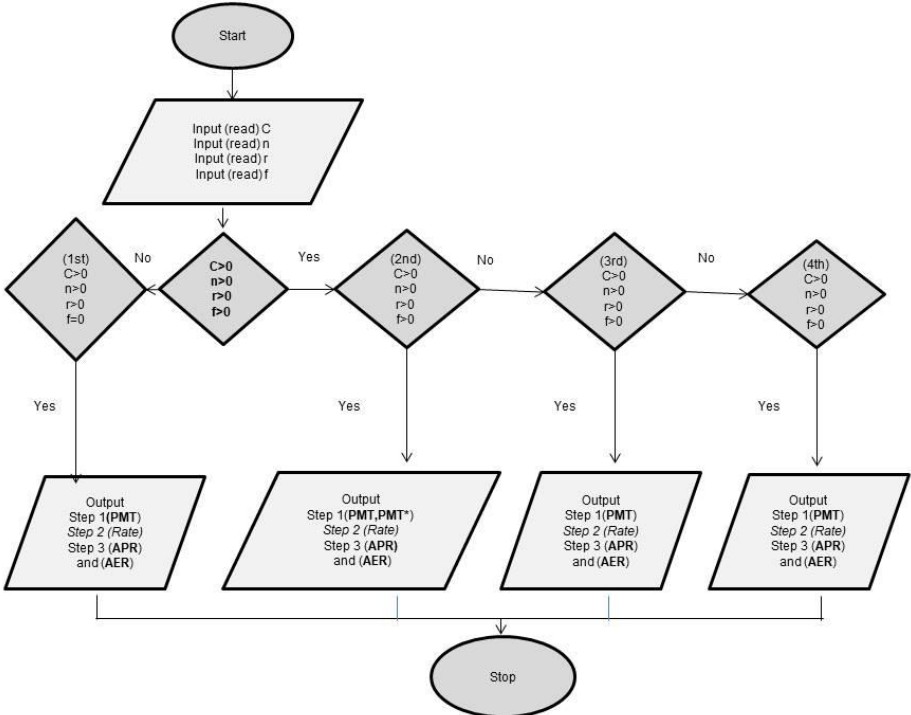

**Figure 1.** The algorithm for a system that supports the transparency of consumer credit offers. Figure 1 Legend. C—Capital, amount of credit/loan. n—number of repayment (usually monthly). r—interest rate (usually annual). f—additional fee (usually given as an interest rate/percentage of amount of credit/loan). PMT—amount of regular repayment of the credit/loans presented in credit offer calculated by usage of the 'PMT' financial function in an Excel sheet or in a financial calculator. * PMT—amount of periodical (monthly) credit/loan repayment. Rate—interest rate per period of repayment of the credit/loans (usually monthly) calculated by usage of the 'Rate' financial function in an Excel sheet or in a financial calculator. APR—Annual Percentage Rate (calculated by multiplying Rate by the number of periods of repayment in the year, e.g., Rate (monthly) × 12). AER—Annual Effective Rate (calculated as effective interest rate, e.g., $(1 + \text{Rate})^{n-1}$).

**Table 3.** Comparison and discussion of consumer credit offers and the results regarding these offers from the system supporting transparency (the amount of credit 20,000 PLN, 48 repayments—PMT constant).

| Case Study No. 1 | | | | |
|---|---|---|---|---|
| Santander offer I | PMT * [PLN] | APR ** | Interest rate | Fee |
| | 487 | 8.13% | 7.84% | 0% |
| *Results from the system supporting the transparency of consumer credit offers* | | | | |
| | 1st case | 2nd case | 3rd case | 4th case |
| amount requested by borrower [PLN] | 20,000 | n/a | n/a | n/a |
| PMT * [PLN] | **487** | n/a | n/a | n/a |
| APR ** | 7.84% | n/a | n/a | n/a |
| AER *** | **8.13%** | n/a | n/a | n/a |

Comment: In the offer from Santander, the PMT is calculated and presented correctly in accordance with standard practice and without additional elements. The APR in the Santander offer is calculated as AER.

**Table 3.** *Cont.*

| Case Study No. 2 | | | | |
|---|---|---|---|---|
| Credit Agricole offer II | PMT * [PLN] | APR ** | Interest rate | Fee |
| | 511.78 | 10.47% | 9.99% | 0% |
| *Results from the system supporting the transparency of consumer credit offers* | | | | |
| | 1st case | 2nd case | 3rd case | 4th case |
| amount requested by borrower [PLN] | 20,000 | n/a | n/a | n/a |
| PMT * [PLN] | **507.16** | n/a | n/a | n/a |
| APR ** | 9.99% | n/a | n/a | n/a |
| AER *** | **10.46%** | n/a | n/a | n/a |

Comment: In the Credit Agricole offer, the PMT is higher than that calculated on the basis of the function PMT and probably includes an additional fee. The PMT = 511.78 PLN is not clear. Taking into account the PMT = 511.78 PLN, the annual APR and AER are 10.47% and 10.99%, respectively.

| Case Study No. 3 | | | | |
|---|---|---|---|---|
| CitiHandlowy offer I | PMT * [PLN] | APR ** | Interest rate | Fee |
| | no data | 9.34% | 7.49% | 2.90% |
| *Results from the system supporting the transparency of consumer credit offers* | | | | |
| | 1st case | 2nd case | 3rd case | 4th case |
| amount requested by borrower [PLN] | n/a | 19,420 (or 20 000) | 20,000 | 20,000 |
| PMT * [PLN] | n/a | 483.48 (or 497.92) | 497.51 | 495.57 |
| APR ** | n/a | 9.02% | 8.98% | 8.78% |
| AER *** | n/a | 9.41% | **9.36%** | 9.14% |

Comment: In CitiHandlowy offer I, there is no possibility for checking the correctness of the PMT calculation (no PMT). The CitiHandlowy offer I APR is calculated as AER and suggests the 3rd case. This means that a fee is added to the amount of credit and partly repaid under each payment with additional interest. The difference between 9.34% and 9.36% may be the result of rounding up/down the payment amount and calculating the last payment adequately in order to repay the whole credit.

| Case Study No. 4 | | | | |
|---|---|---|---|---|
| CitiHandlowy offer II | PMT * [PLN] | APR ** | Interest rate | Fee |
| | 500.75 | 9.32% | 7.49% | 2.90% |
| *Results from the system supporting the transparency of consumer credit offers* | | | | |
| | 1st case | 2nd case | 3rd case | 4th case |
| amount requested by borrower [PLN] | n/a | 19,420 (or 20,000) | 20,000 | 20,000 |
| PMT * [PLN] | n/a | 483.48 (or 497.92) | 497.51 | 495.57 |
| APR ** | n/a | 9.02% | 8.98% | 8.78% |
| AER *** | n/a | 9.41% | 9.36% | 9.14% |

Comment: In CitiHandlowy offer II, the PMT = 500.75 PLN, which suggests that it includes an additional fee (instead of 2.9%).
The PMT = 500.75 PLN is not clear. Taking into account the PMT = 500.75 PLN, the annual APR and AER are 9.32% and 9.73% respectively.

**Table 3.** *Cont.*

| Case Study No. 5 | | | | |
| --- | --- | --- | --- | --- |
| Alior Bank offer I | PMT* [PLN] | APR** | Interest rate | Fee |
| | 536 | 13.76% | 7.49% | 10.90% |
| *Results from the system supporting the transparency of consumer credit offers* | | | | |
| | 1st case | 2nd case | 3rd case | 4th case |
| amount requested by borrower [PLN] | n/a | 17,820 (or 20,000) | 20,000 | 20,000 |
| PMT * [PLN] | n/a | 483.48 (or 542.63) | **536** | 528.9 |
| APR ** | n/a | 13.61% | 12.96% | 12.23% |
| AER *** | n/a | 14.49% | **13.76%** | 12.94% |

Comment: In Alior Bank offer I, the PMT is calculated and presented correctly in accordance with standard practice and without additional elements. The APR in Alior Bank offer I is calculated as AER and it suggests the 3rd case. This means that a fee is added to the amount of credit and partly repaid under each payment with additional interest.

| Case Study No. 6 | | | | |
| --- | --- | --- | --- | --- |
| Alior Bank offer II | PMT * [PLN] | APR ** | Interest rate | Fee |
| | 511.1 | 10.40% | 9.50% | 6.90% |
| *Results from the system supporting the transparency of consumer credit offers* | | | | |
| | 1st case | 2nd case | 3rd case | 4th case |
| amount requested by borrower [PLN] | n/a | 18,620 (or 20,000) | 20,000 | 20,000 |
| PMT * [PLN] | n/a | 502.46 (or 539.70) | 537.13 | 531.21 |
| APR ** | n/a | 13.32% | 13.06% | 12.46% |
| AER *** | n/a | 14.16% | 13.87% | 13.20% |

Comment: In Alior Bank offer II, the way of calculating PMT = 511.10 PLN is not clear. Probably only part of the fee presented in the offer (6.90%) is taken into account. Taking into account the PMT = 511.1 PLN, the annual APR and AER are 10.40% and 10.91% respectively.

| Case Study No. 7 | | | | |
| --- | --- | --- | --- | --- |
| PNB Paribas offer I | PMT * [PLN] | APR ** | Interest rate | Fee |
| | 538 | 13.92% | 9.99% | 6.00% |
| *Results from the system supporting the transparency of consumer credit offers* | | | | |
| | 1st case | 2nd case | 3rd case | 4th case |
| amount requested by borrower [PLN] | n/a | 18,800 (or 20 000) | 20,000 | 20,000 |
| PMT * [PLN] | n/a | 507.16 (or 539.53) | **537.58** | 532.16 |
| APR ** | n/a | 13.30% | 13.10% | 12.56% |
| AER *** | n/a | 14.14% | **13.92%** | 13.30% |

Comment: In PNB Paribas offer I, the PMT is calculated and presented correctly in accordance with standard practice and without additional elements. The APR in PNB Paribas offer I is calculated as AER, and it suggests the 3rd case. This means that a fee is added to the amount of credit and partly repaid under each payment with additional interest.

**Table 3.** *Cont.*

| Case Study No. 8 | | | | |
|---|---|---|---|---|
| PNB Paribas offer II | PMT * [PLN] | APR ** | Interest rate | Fee |
| | 514 | 10.70% | 6.99% | 9.90% |

*Results from the system supporting the transparency of consumer credit offers*

| | 1st case | 2nd case | 3rd case | 4th case |
|---|---|---|---|---|
| amount requested by borrower [PLN] | n/a | 18,020 (or 20,000) | 20,000 | 20,000 |
| PMT * [PLN] | n/a | 478.83 (or 531.45) | 526.24 | 520.08 |
| APR ** | n/a | 12.48% | 11.96% | 11.33% |
| AER *** | n/a | 13.22% | 12.63% | 11.93% |

Comment: In PNB Paribas offer II, the way of calculating PMT = 514 PLN is not clear. Probably only part of the fee presented in the offer (9.90%) is taken into account. The trial and error method allows for supposition that to calculate the PMT = 514 PLN a fee of 8.4% was used (300 PLN was probably ignored in the calculation of PMT and may have been paid in another way) and suggests the 4th case. This means that a fee is added to the amount of credit and partly repaid under each payment without additional interest. Taking into account the PMT = 514 PLN, the annual APR and AER are 10.7% and 11.2%, respectively.

| Case Study No. 9 | | | | |
|---|---|---|---|---|
| HSBC UK | PMT * [GDP] | APR ** | Interest rate | Fee |
| | 474,32 | 6.7% | 6.7% | 0% |

*Results from the system supporting the transparency of consumer credit offers*

| | 1st case | 2nd case | 3rd case | 4th case |
|---|---|---|---|---|
| amount requested by borrower [GBP] | 20,000 | n/a | n/a | n/a |
| PMT * [GBP] | 474,32 | n/a | n/a | n/a |
| APR ** | 6.5% | n/a | n/a | n/a |
| AER *** | 6.7% | n/a | n/a | n/a |

Comment in the offer from HSBC UK: Taking into account the given PMT, the APR in the HSBC UK offer is calculated as AER. The parameter "Interest rate of 6.7% p.a." is not clear.

| Case Study No. 10 | | | | |
|---|---|---|---|---|
| Barclays Bank | PMT * [GDP] | APR ** | Interest rate | Fee |
| | 475.19 | 6.8% | 6.8% | 0% |

*Results from the system supporting the transparency of consumer credit offers*

| | 1st case | 2nd case | 3rd case | 4th case |
|---|---|---|---|---|
| amount requested by borrower [GBP] | 20,000 | n/a | n/a | n/a |
| PMT * [GBP] | 474,32 | n/a | n/a | n/a |
| APR ** | 6.6% | n/a | n/a | n/a |
| AER *** | 6.8% | n/a | n/a | n/a |

Comment in the Barclays Bank offer: Taking into account the given PMT, the APR in the Barclays Bank offer is calculated as AER. The parameter "Interest rate of 6.8% p.a." is not clear.

* PMT—amount of periodical (monthly) credit/loan repayment. ** APR—annual percentage rate. *** AER—annual effective rate. Source: Tables 1 and 2, and own calculations on the basis of the algorithm in Figure 1.

The results of the analysis presented in Table 3 provide an answer to research question no 2. The presentation of credit offer costs does not use a standardized parameter APR calculated in the same way in every case. This makes it impossible for credit offers to be properly compared.

The document analysis on a consumer credit offer with results of the system supporting the transparency of consumer credit offers reveals many irregularities:

1.　Some documents in the form of credit offers are inaccurate and misleading.
2.　Some credit offers declare that the interest is the only cost but despite this include additional costs.
3.　Many credit offers do not distinguish between the APR and AER and confuse the two concepts.
4.　The amount of PMT included in the offers is not always clear and understandable, taking into account the given parameters (amount of credit, numbers of repayments, interest rate, fee).
5.　In many credit offers, it is difficult to determine the way the fee is charged due to inappropriate calculations and/or presentation of the elements of financial offers regarding consumer credit.
6.　Some credit offers ignore or add part of the fees to the calculation and in the presentation of important elements of the offers (i.e., PMT, APR, and AER).

In the analysis presented in Table 3, the algorithm presented in Figure 1 was used. The use of the algorithm, which was developed and dedicated to credit consumers and which may be used when making a credit decision, provides an answer to research question no. 3 and confirms that it is possible to develop a system to support and increase the transparency of consumer credit, which also increases consumer protection.

The system developed on the basis of the algorithm presented in Figure 1, which supports the transparency of consumer credit offers, provides not only correct calculations of PMT, APR, and AER but can also detect errors in APR and AER calculations if there are any. In addition, the system fulfills an educational function, making consumers more aware of credit in terms of how the way fees are charged influences the APR and AER.

The results of the analysis exemplify the existing and as yet unresolved problems of the lack of transparency of information in the presentation of financial offers. In the case of consumer credit offers, an additional problem is mistakes (intentional or accidental) in the APR calculation. This confirms the need to develop different tools (rules and support systems) to facilitate informed credit decision-making.

## 5. Conclusions

The objectives of the research, both theoretical and practical, were achieved. The document analysis conducted confirms that many consumer credit offers are not presented in a transparent and clear way. The presentation of credit offer costs does not always use a standardized APR parameter calculated in the same way in every case. This makes it impossible for many credit offers to be properly compared.

Recognizing the inaccurate and misleading information in credit offers allowed a wider perspective on these mistakes to be captured. This may be used to develop relevant solutions of responsible lending regimes, their supervisory activities, and their financial education and awareness solutions in the field of consumer credit.

It turns out that even consumer credit regulations with greater constraints designed to prevent consumer credit price manipulation are not enough. What is necessary is a change in thinking toward money and finance that follows a more humanistic and ethical path, with a simultaneous increase in meaningful education and sound financial practices. In addition, practical solutions in the form of various tools/systems to support the transparency of consumer credit offers are also required.

The proposed system supporting the transparency of consumer credit offers helps to raise awareness and exposes information that is not transparent and is unclear, which is usually the basis for credit decision-making. The system presents standardized parameters (PMT, APR, and AER), which make it possible to effectively compare credit offers. In addition, the system uses a correct and highly popular method for calculating the PMT on the basis of the PMT function (rate, nper, pv,

[fv], [type]). In turn, the method for calculating the APR and AER uses the RATE function (nper, pmt, pv, [fv], [type], [guess]) to indirectly calculate the periodic effective interest rate (r). The system fulfills an educational function and provides a correctly calculated set of parameters that facilitates financial decision-making and raises consumer awareness. This means that the system increases financial protection for consumers. The simple version of the system supporting the transparency of consumer credit offers presented here may be developed as necessary for more complicated cases.

**Funding:** This research received no external funding.

**Conflicts of Interest:** The author declares no conflict of interest.

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
