# Peer review of "A System to Support the Transparency of Consumer Credit Offers"

_jrfm, doi:10.3390/jrfm13120317_

Round 1

Reviewer 1 Report

The paper is intended to be a qualitative analysis of a set of consumer credit documents.

In the Abstract the author mentions that `The system developed uses a computer to simulate human thinking and to augment it with artificial intelligence` but I did not understand how it was developed and used.

The Introduction is too long (3 pages) and does not describe the objectives of the study. The objectives and research questions are presented in Methods section - Line 118 `The theoretical objective of the paper is to examine the completeness of the information` and Lines 120-121 `The empirical objective is the development of conclusions and implications` - but not the research methods.

It is necessary to revise the Introduction to reflect the aims of the paper, the relevance of the research and methodology.

The Literature review is missing – there are a few references in the Introduction. I recommend expanding the literature references and including in a separate section.

The section 3 - `Annual percentage rate as an important criterion in credit decision-making` - is a short presentation of APR not related to the objectives of the study and the research questions.

The Analysis and Discussion must be improved. Some questions that this section should answer: Do the results of the analysis support or contradict existing theories? What are the relevance and implications of the paper results?

The Conclusions are not complete, please revise this section.

Author Response

I hereby confirm that the paper titled

A system to support the transparency of consumer credit offers

is a product of my own work.

The results of the research presented in this article show the example of solving the problem of consumer protection by the  making conscious decisions (e.g. in borrowing money). A significant problem in making conscious decisions is the lack of information or unclear information regarding financial offers, including consumer credit. Financial protection for consumers can be increased by using systems that support consumers faced with a lack of transparency of consumer credit offers. This may facilitate practices for informing consumers about the characteristics of consumer credit, and at the same time support the implementation of the concept of responsible lending. The system developed in this article uses a computer to simulate human thinking and to augment it with artificial intelligence. It facilitates the elimination the behavioral biases during the taking of financial decisions, which are the result of a low level of financial literacy.

Dear Reviewer,

The author wishes to thank you for your constructive comments and suggestions regarding the article.

Main changes

  1. Comment 1

“In the Abstract the author mentions that `The system developed uses a computer to simulate human thinking and to augment it with artificial intelligence` but I did not understand how it was developed and used.”

Answer:

The artificial intelligence in the system to support the transparency of consumer credit offers, which is built on the basis of an algorithm, entails the use of computers or other tools for problem-solving (the problem regards supplementary information as well as the calculation the parameter - APR or/and AER - which is calculated in the same way).

  1. Comment 2

“The Introduction is too long (3 pages) and does not describe the objectives of the study. The objectives and research questions are presented in Methods section - Line 118 `The theoretical objective of the paper is to examine the completeness of the information` and Lines 120-121 `The empirical objective is the development of conclusions and implications` - but not the research methods. It is necessary to revise the Introduction to reflect the aims of the paper, the relevance of the research and methodology.”

Answer:

The introduction has been revised and shortened to 2 pages, and the objectives have been moved from the methodology section to the introduction.

The research method is presented in the next section.

  1. Comment 3

“The Literature review is missing – there are a few references in the Introduction. I recommend expanding the literature references and including in a separate section.”

Answer:

The literature review has been expanded and confirms the consideration included in the introduction regarding the research gap (i.e. the missing kinds of research on APR and AER) and formulating the objectives.

An additional section for the literature review has not been specified due to suggested sections of the Journal of Risk and Financial Management: Introduction (with literature review), Methods, Results, etc.

  1. Comment 4

“The section 3 - `Annual percentage rate as an important criterion in credit decision-making` - is a short presentation of APR not related to the objectives of the study and the research questions”.

Section 3 - Annual percentage rate as an important criterion in credit decision-making` - is a short presentation of APR and AER and is directly related to the algorithm created as part of the system, which is the empirical objective of the study and which is related to research question no 3. (Is it possible to develop a system to support and increase the transparency of consumer credit, which also increases consumer protection?)

  1. Comment 5

“The Analysis and Discussion must be improved. Some questions that this section should answer: Do the results of the analysis support or contradict existing theories? What are the relevance and implications of the paper results?”

Answer:

The Analysis and Discussion has been improved. The discussion section has been expanded, mainly by underlining the response to the research questions. In addition, it includes information that the results of the analysis confirm the existing and as yet unresolved problems of the lack of transparency of information in the presentation of financial offers, while at the same time proposing a solution in the form of the system to support the transparency of consumer credit offers.

  1. Comment 6

“The Conclusions are not complete, please revise this section.”

Answer:

The Conclusions have been revised. This took into account the main conclusions from the document analysis and the main implications regarding the suggested directions in changes in personal finance - mainly more humanistic and ethical thinking with regard to consumer finance, financial education and additional practical solutions to support financial decision-making.

  1. All new changes to the text are highlighted in green.
  2. The writing has been corrected and a professional proofreading service was used.

Reviewer 2 Report

The manuscript is devoted to an important issue of supporting the transparency of consumer credit offers. The subject matter discussed in the article is an interesting one, and timely in the era of dynamically occurring changes on the global market. The issue, although it cannot be considered as fully novelty, it is however useful from the scientific point of view and interesting for the readers. The considerations are conducted in a correct and logical manner.

The statement is transparent, factually correct and well structured. The cited bibliography confirms the thesis put forward by the authors. Data sources in figures and tables clearly indicate where the data comes from or who is their author. Numerous tabular statements are sufficiently commented. The presented final conclusions sufficiently relate to the thesis put forward by the authors. The English language and style of the presented issue are fine.

The above proves that the manuscript can be recommended for publication.

Author Response

The Cover Letter

I hereby confirm that the paper titled

A system to support the transparency of consumer credit offers

is a product of my own work.

The results of the research presented in this article show the example of solving the problem of consumer protection by the  making conscious decisions (e.g. in borrowing money). A significant problem in making conscious decisions is the lack of information or unclear information regarding financial offers, including consumer credit. Financial protection for consumers can be increased by using systems that support consumers faced with a lack of transparency of consumer credit offers. This may facilitate practices for informing consumers about the characteristics of consumer credit, and at the same time support the implementation of the concept of responsible lending. The system developed in this article uses a computer to simulate human thinking and to augment it with artificial intelligence. It facilitates the elimination the behavioral biases during the taking of financial decisions, which are the result of a low level of financial literacy.

Dear Reviewer,

The author wish to thank for the review, which accepts the paper

According to the comments and suggestion of second review a few changes have been conducted.

Round 2

Reviewer 1 Report

I think that the manuscript has been improved by the author` revisions.